# Building Energy Use: Modeling and Analysis of Lighting Systems—A Case Study

**Aron Powers * and Messiha Saad ***

School of Engineering & Applied Sciences, Washington State University, 2710 Crimson Way, Richland, WA 99354, USA

* Correspondence: aron.powers@wsu.edu (A.P.); messiha.saad@wsu.edu (M.S.)

**Abstract:** Understanding how energy is used and where it can be saved in an existing building is critical not only from a cost and environmental standpoint, but for legal compliance as well, as the United States and the rest of the world increasingly have set tighter restrictions on energy usage and carbon emissions. Energy savings can be achieved from installing LED lights and occupancy sensors; however, the exact savings and impact of each method can vary depending on the building in question. The objective of this case study is to perform analysis of the lighting systems in Washington State University Tri-Cities' Floyd & East buildings to determine energy savings potential. Lighting systems in each building were broken into several groups based on their operational patterns and then numerically modeled with the aid of Python. The results of this case study shows that 60% energy savings, totaling 350 MWh in a year, can be achieved by retrofitting fluorescent lights with LEDs and occupancy sensors. This energy savings translates to a reduction of 62.4 t of $CO_2$ emissions per year. The results of our cost-analysis in this model shows that the LED light retrofit has a break-even point at 15 months of operation.

**Keywords:** energy analysis; lighting; energy modeling; building energy; university; energy retrofit





## 1. Introduction

### 1.1. Relevance of Building Energy Analysis

Reducing the energy usage in buildings is of great importance. According to the U.S. Energy Information Administration (Washington, DC, USA), commercial, residential , and industrial buildings accounted for nearly three-quarters of all energy consumption in the United States during 2020, with the remainder being used for transportation [1]. The major consumers of electricity inside facilities were lighting, refrigeration, ventilation, cooling, and computer systems, totaling an approximate use of 913 billion kWh [2]. With nearly four-fifths of all energy generation coming from carbon-based fuels in the United States [1], it can be reasoned that the majority of energy used in buildings directly contributes to greenhouse gas emissions either through the usage of electricity that relies on carbon-based fuels for generation or through the direct utilization of these fuels.

While it is possible to change the methods of electrical generation in the United States, it is much more simple to reduce the usage of electricity downstream. The large cost to design, build, and maintain generation capacity outweighs the benefits of adding capacity. Additionally, it does not solve the root problem: too much energy usage. Excessive energy usage leads to increased fossil fuel usage. This naturally leads to increased emissions, exacerbating global warming effects. Rather than increase the energy produced, consumers should be employing more efficient versions of the equipment they operate.

Energy efficient building systems are not strictly limited to new buildings, as great energy savings can be realized in existing infrastructure. Xing, Hewitt, and Griffiths estimate that by the year 2050, 80% of the buildings humans will occupy will have already been built [3]; therefore, existing buildings are where the greatest capacity for energy savings

lie [3]. Given this information, it is more important than ever that building managers look at ways to reduce energy usage in the buildings they currently operate.

*1.2. Impact of Washington State Law*

As a response to the current energy market, some state and federal legislatures have taken action to reduce energy consumption. One such state to adopt new guidelines is the state of Washington. On 17 January 2019, HB 1257 was introduced to the Washington State Legislature, commonly referred to as the Clean Buildings bill [4]. It sought to set limits on covered commercial building energy usage for facilities of 50,000 gross square feet and over, citing improved air quality, reduced capital spending on power generation, lower building operating costs, and reducing greenhouse emissions as motivations for the bill [4]. The bill uses ASHRAE Standard 100-2018 as a base, specifying several energy use intensity targets ($EUI_t$) for each type of building affected [4].

The ASHRAE standard that powers the bill includes three important dates for compliance in non-residential commercial structures. Buildings bigger than 220,000 square feet must comply by 1 June 2026 [5]. For buildings more than 90,000 square feet but less than 220,001 square feet, the compliance date is moved out a year to 1 June 2027 [5]. Finally, buildings that have less than 90,001 square feet but more than 50,000 square feet are required to comply by the year after, 1 June 2028 [5]. For early adopters, incentives are available of $0.85 per square foot minimum are available [5].

One way in which building owners can be in compliance with the law is through exemption. According to Normative Annex Z4.1 of the Washington State Clean Buildings Performance Standard, the building must meet one of the following criteria [5]:

1.  The building did not have a certificate of occupancy for 12 consecutive months within a 2 year period prior to the compliance date.
2.  The building was not physically occupied by either the owner or tenant in no less than 50% of the conditioned floor area for the 12 consecutive months prior to the compliance date.
3.  More than half of the gross floor area is used for manufacturing activities in Group F or Group H as specified in the Washington State Building Code.
4.  The building is for agricultural use.
5.  The building is pending demolition.
6.  The building has undergone financial hardship.

If the exemption is submitted but not approved, the building owner must then begin to determine if their building is to comply with the $EUI_t$ guidelines or through investment criteria [5]. If the $EUI_t$ for a building is measurable and there is a defined target for the building type in question, the $EUI_t$ path must be taken [5]. To determine the building's EUI (Energy Usage Intensity), the net energy consumption is divided by the gross floor area of the building [5]. Once this is done, the $EUI_t$ is determined by a weighting of the floor area by specific factors defined in the standard. In order to comply with the law, the calculated EUI must not exceed the $EUI_t$ determined previously [5].

In the event that the EUI is not measurable for a building or that is does not have a specific target listed in the law, the final path to compliance is through investment [5]. In order to comply through this path, the building owner must "complete a LCCA (life-cycle cost analysis) and implement an optimized bundle of energy efficiency measures that provide maximum energy savings without resulting in a savings-to-investment ratio of less than one" [5].

In addition to the above compliance requirements, building owners are also required to submit plans detailing operations and maintenance of building systems as well as an energy management plan [5]. Building owners are to come up with plans to maintain the systems in the building envelope, building systems, and building equipment that consume energy to ensure efficient operation over the useful lifespan of the systems [5].

### 1.3. Lighting Systems

The earliest forms of artificial light employed by humans were the lights of campfires and torches. Primitive oil lamps were also available that utilized animal and vegetable fats. Candles for general illumination were also common. It was not until the discovery of whale oil in the mid 1800's that lighting technology began to rapidly transform [6]. Kerosene production overtook whale oil as the primary fuel source for lanterns and dominated the majority of light fixtures. This was the result of kerosene being able to brighter and be manufactured more economically. Once electricity became more ubiquitous, arc lights and incandescent bulbs arose, replacing fuel-based lights for general indoor and outdoor activities. In the modern day, most lights used are fluorescent lights as well as other discharge lamps like metal halide and high pressure sodium lamps; however, LED lights are emerging to the forefront as the next replacement. LED lights are more efficient, can easily emit a variety of colors, are easily dimmable through pulse-width modulation, generally last longer, and can be suited for small or large applications. In addition, occupancy sensors are also another modern invention to control lighting in an efficient manner, while light is necessary to see, leaving the lights on when they are not needed leads to waste. Occupancy sensors seek to solve this problem by switching the lights on whenever occupants are in the room and off when they leave. This is done via (1) ultrasonic sensors, (2) measuring environmental changes in the room, (3) infrared sensors, and (4) other activation methods.

A technical definition for light comes from The Society of Light and Lighting and is defined as "part of the electromagnetic spectrum that stretches from cosmic rays to radio waves" [7]. Human photoreceptors are only able to absorb electromagnetic waves in the 380–780 nm range, so this range of light is defined as visible light [7]. Light's most fundamental measure for electromagnetic radiation is radiant flux, defined to be the rate of flow of energy emitted from an electromagnetic wave, and in the SI system is measured with the watt [7]. Light in the visible spectrum is classified by luminous flux (lumens). For the visible spectrum, it is calculated by the following product:

$$\Phi = K_m \sum_i \Psi_i V_i \Delta\lambda \tag{1}$$

where $\Phi$ is the luminous flux in lumens, $K_m$ is a constant based on the definition of the observer, $\Psi_i$ is the radiant flux in a wavelength interval $\Delta\lambda$, and $V_i$ is the relative luminous efficiency function based on the light condition [7].

The final important measure for light is illuminance, defined to be the amount of luminous flux per area illuminated (lumens/m$^2$ in the SI system, named lux). Spaces with activities that involve detailed work require higher illuminance than spaces with less-detailed work, and illuminance is a way to quantify this independently of workplane area since illuminance is effectively a light density. Standards define space lighting requirements via illuminance as it allows for direct comparisons of spaces irregardless of their size. For field verification purposes, illuminance meters are employed to measure a space's illuminance in lux. On a basic level, illuminance meters consist of a photodiode and a photopic correction filter [7]. Through the appropriate signal processing components (i.e., operational amplifiers, filters, microprocessors), the measured lux is then displayed to an LCD screen. Illuminance meters come in a range of form factors, from simple handheld devices all the way to rack-mounted laboratory-grade equipment [7].

Literature has attempted to identify ways in which lighting energy can be reduced. In Nagy et al., it was reported that with proper calibration, occupancy sensors can be employed to reduce the power consumption in lighting fixtures while maintaining comfortable lighting levels, obtaining from 23.2% to 37.9% energy savings in typical office spaces [8]. Previous literature reported similar reductions in power consumption, with Reinhart observing 20% to 60% reductions with combinations of occupancy sensing and automatic blinds control [9].

Labeodan et al. investigated the effects of occupancy control in lighting systems and demonstrated two important findings. The first was that energy savings were on average

24% as compared to normal operations, and the second being that the first could be done with satisfactory occupant comfort [10], while occupants noted distractions to their day-to-day work arising from the operation of occupancy sensors in the first week of the study, a combination of becoming used to their operation and tweaks to the luminaire switching interval were cited for being the reason behind increasing user satisfaction as the weeks of the study progressed [10]. It should also be noted that the study proposed that occupancy sensing was not a compelling investment based on the rate of return from the reduced electricity usage, but it is still a way to increase the energy performance nevertheless [10].

South African schools were able to save 21–39% of their electric energy costs by retrofitting fluorescent fixtures with LED tubes as reported by Booysen, Samuels, Grobbelaar [11]. In the article, 60 W fluorescent lamps were compared to various LED lamp offerings, and a model was created to rapidly assess the benefits from retrofitting the school with LED lamps [11]. The authors' long-term projections found that "the most expensive [LED] light in the short them yielded the most savings in the long term, and that the price tag and long-term savings do not necessarily correspond to light quality", demonstrating that the higher costs of quality LED lights may be deceiving when the entire life cycle cost is considered [11].

LED lighting as a technology has made significant improvements in being a viable alternative to traditional discharge lamps like fluorescent, metal halide, and high pressure sodium lights; however, heat dissipation remains a key factor to bulb life. Using a traditional Galerkin-method finite element analysis in COMSOL, Hamida and Hatami were able to design and optimize a heatsink for the cooling of the circuitry in LED lights [12]. Their work demonstrated that using $Al_2O_3$-water as a working fluid and through the usage of thin, long fins, the nanofluid temperature can be increased by up to 6.5% through the design optimization, enhancing the LED package to resist heat better [12]. Work from Hamida, Almeshaal, and Hajlaoui add to the improved thermal regulation of LED packages by using COMSOL to model the junction temperature [13]. Similar to their previous work, the LED package is attached to a heatsink and nanofluid motion through the heat sink is modeled via microchannel flow [13]. It was demonstrated that MWCNT-Water was able to promote a 8.3% reduction in temperature of the junction when compared to MWCNT-Ethylene Glycol and a 14.9% for MWCNT-Engine Oil [13]. In addition to these studies, several other papers have published demonstrating the advancements in LED component cooling [14–17]. Other improvements to LED lights were noted in Danyali and Moteiri where a photovoltaic LED light setup is driven by a Zeta-Sepic converter, managing the LED light to run at peak efficiency [18]. Their system considered the optimization of light emission, electrical, and thermal energy in the design, validating their results both with a numerical simulation in MATLAB and experimentally [18]. The control system was able to work well to balance each consideration and provide an efficient LED package for street lighting [18].

Exterior lights have also been a field of heavy study, as many of the lights that illuminate walkways and entryways are considerably higher power than interior lights. Gorgulu and Kocabey investigated the energy-savings potential of exterior lights on a Turkish campus by calculating the energy consumption of several different scenarios and found that a combination of strategies; namely, dimming, LED retrofitting, and occupancy timing, could save up to 762 MWh of energy and €1,300,000 over 12 years [19]. Another case they considered was to simply turn the exterior lights off 30 min after foot traffic should cease, yielding the second-largest $CO_2$ emissions and energy savings; however, they noted the obvious negative safety implications for campus security at night [19]. They also note that while the LED retrofits do provide a return on investment, they also require a significant upfront cost compared to traditional high pressure sodium and metal halide lights [19]. In 2015, an article by Tähkämö, Räsänen, and Halonen investigated the life cycle costs of high-pressure sodium and LED bulbs in Finland for street lighting, citing the European Union ban of high-pressure mercury bulbs as the driving factor behind the analysis [20]. Using 30 years as the time frame, a combination of part purchase prices, freight charges,

and installation fees were considered in the initial investment cost in addition to operational and maintenance fees associated with each luminaire [20]. In the end, it was found that the LED bulbs were overall less economically viable; however, they also noted that the price is expected to continuously drop and that LED lights can offer better lighting controls and more favorable light color rendering [20].

A energy retrofit case study was performed on the Santi Romano Dormitory on the Palermo University campus by Curto, Franzitta, Guercio, and Panno where thermal insulation, mechanical/thermal systems, and lighting were analyzed [21]. In their study, the dormitory was first audited for its relevant features and subsequently analyzed through models created in Excel [21]. The insulation losses, water heating load, summer cooling load, and lighting loads were generated [21]. Once the initial loading was generated, subsequent retrofits were suggested and then analyzed such as window replacements, sunscreen films, and chiller and boiler replacements [21]. In total, 514,477 kWh/year (65%) of electrical energy savings and 482,671 kWh/year (33%) of thermal energy savings were possible through the retrofits described in their analysis [21].

*1.4. Purpose*

Washington State University Tri-Cities (WSUTC) is located in Richland, Washington in The United States of America. The Floyd & East buildings on campus are considered one building for the purposes of determining gross floor area due to the presence of a thermally-insulated walkway between them. Floyd was constructed in 1991, and East was constructed in 1968. The total thermally-insulated square footage of the buildings are approximately 134,000 square feet, thus subjecting the structure to HB 1257. This paper seeks to perform an energy retrofit case study on the Floyd and East buildings at Washington State University Tri-Cities, numerically modeling energy consumption and exploring retrofit options for lighting systems based on the existing lighting infrastructure. Once modeled, the results are to be analyzed and summarized, providing vital information for Washington State University Tri-Cities in terms of direct energy savings, emissions reductions, and a cost-analysis for changing fluorescent lights with LEDs. Additionally, the results will serve as an example for similar United States retrofits where the majority of literature is based overseas.

## 2. Methodology

In analyzing the energy consumed by lights, it is necessary to break each space into groups to allow for the assumptions of each group to better represent the actual conditions in the space. In each group, a base case is calculated and then compared to successive retrofit improvements as applicable. There are four classifications presented:

1. Continually-operated interior lights representative of hallways, bathrooms, stairwells, and common areas
2. Exterior lights for walkways, parking lots, and night lighting.
3. Classroom/lab space lighting, specifically where students have regularly-scheduled classes.
4. Office spaces and miscellaneous spaces.

In each group, all fluorescent lights are 4 foot long F32T8/TL835 PLUS ALTO by Philips operating under an instant-start ballast. This comes from both a visual inspection of lights on campus and by speaking with facilities maintenance. In the retrofit from fluorescent to LED tubes, a MAS LEDtube VLE UN 1200 mm UO 15.5W840 T8 by Philips was chosen as it saves a significant amount of energy, has a similar nominal lumen output to the F32T8/TL835 PLUS ALTO fluorescent tube, and can be installed either directly to mains or placed in an existing luminaire with a ballast. The LED bulbs can be installed directly to any of the existing fixtures; therefore, the only retrofit effort required is to replace the bulb. Table 1 shows technical specifications for the lights [22,23]. For the lumen output on the fluorescent lights, the design mean lumens is used, and 12-h instant start is used as the operating characteristic for the bulb life.

**Table 1.** Fluorescent and retrofit LED tube specifications adapted from Philips Lighting.

| Light Name | Power | lm | lm/W | Light Color | Bulb Life |
|---|---|---|---|---|---|
| F32T8/TL835 PLUS ALTO 30PK | 32 W | 2800 | 87.5 | 3500 K | 38,000 h |
| MAS LEDtube VLE UN 1200 mm UO 15.5W840 T8 | 15.5 W | 2500 | 161 | 4000 K | 60,000 h |

The building contains 10 W LED canister lights. These will be denoted as LED canisters and are not to be confused with the LED lights installed for retrofitting fluorescent lights. Any high intensity discharge (HID) lights are 250 W lights unless otherwise noted. The energy consumption of the occupancy sensors and fluorescent ballasts are neglected in this model.

*2.1. Continuous Operation*

There are sections of the building where the lights operate on a near continuous basis. These areas include hallways, stairwells, and bathrooms. During field data collection and speaking with occupants of the building, these lights are on nearly 24/7 and, lacking user intervention for off-normal events, will continue to be on 24/7. For the purpose of analysis in these spaces, it may be reasonably assumed that the electric load for a typical day is constant and that the lights remain on 24/7.

Four scenarios are modeled in this category. (1) The first case assumes no retrofit, serving as a baseline model for which the other cases can be compared to. (2) The second case is to replace all fluorescent lights with LED equivalents, representing a drop-in replacement scenario. (3) The third case is to assume only occupancy sensing is added, creating a benchmark to analyze any potential benefits of replacing lights versus adding occupancy sensing. (4) Finally, the fourth case is to combine both LED retrofitting and occupancy sensing.

First, the power for each scenario is generated. For each type of light in the group, the power is defined to be:

$$P = \sum_{i=1}^{3} n_i p_i \tag{2}$$

where $P$ is the total power, $n_i$ is the number of lights for bulb $i$, and $p_i$ is the power of bulb $i$. Since there are three light fixtures considered, $i = 1, 2, 3$. For the base case and the LED retrofit, Equation (2) works without modification to obtain the power used at each hour since as there is no variation in loading to consider. In considering the occupancy sensors and the LED retrofit plus occupancy sensors, a modification to Equation (2) is required to account for light switching. Either by code or policy, lights in this category will not completely turn off. Approximately one third of the fluorescent lights present in hallways and stairwells in the continuous operation group stay on to provide egress lighting. Bathroom lights are equipped with a phosphorescent tube around the exterior of the fixture to provide emergency egress lighting, so they are not considered. Since the normal operating hours are from 7 AM to 10 PM, the lights are considered to be fully on during this time and follow Equation (2). To account for the lights that remain on during egress conditions, Equation (2) is modified to be:

$$P = \sum_{i=1}^{3} n_i p_i f_i \tag{3}$$

where $f_i$ represents the fraction of lights that remain on in off-periods. As mentioned above, $f_i = \frac{1}{3}$ for fluorescent lights in this category. Once the power is obtained, then the following expression can be evaluated to find the total energy consumption:

$$E = \int_{t_a}^{t_b} P \, \mathrm{d}t \tag{4}$$

where $t_a$ is the time the lights are turned on in hours, $t_b$ is the time where the lights are turned off in hours, and $E$ is the total energy consumed from $t_a$ to $t_b$. This model, and all other models presented in this paper, use a discrete time step of $\Delta t = \frac{1}{12}$ h (5 min) and use numerical integration to carry out the integration over 24 h. All points are to be connected by a straight line; therefore, the integration can be reduced to a trapezoid rule.

### 2.2. Exterior Lighting

On the exterior of the Floyd & East buildings, there are many HID lamps illuminating the sidewalks, entryways, and parking lots. In the parking lots, 250 W metal halide lamps are used, and 100 W high pressure sodium lamps are employed on the sidewalks, walkways, doorways, and in any non-parking applications. For purposes of comparison, the 100 W high pressure sodium lights are modeled using the characteristics of a Philips Ceramalux ALTO SON 100 E39 ED75 CL SL/12, as the bulbs listed are similar in style and function to the lamps present. The 250 W metal halide bulbs are assumed to be a Philips Switch Start Metal Halide Standard MH250/U UPC 3PK for the same reasons as the 100 W bulbs. When swapping to LED bulbs, the Philips LED Wallpack 60WP/LED/840/ND EX39 G2 BB 6/1 bulb is used for retrofitting the 100 W high pressure sodium lights, and the Philips LED Corncob 150CC/LED/850/ND EX39 G2 BB 3/1 is used to retrofit the 250 W metal halide lamps. In a similar fashion to the high pressure sodium and metal halide lamps, the LED lamps were chosen for their extremely high efficiencies, much longer life, and similar lumen output. Table 2 shows a summary of the technical specifications of these bulbs [24–28].

**Table 2.** Exterior light tube specification sheet data adapted from Philips Lighting data.

| Category | Light Name | Power | Lum. Flux | lm/W | Light Color | Bulb Life |
|---|---|---|---|---|---|---|
| Walkway | SON 100 E39 ED75 CL SL/12 | 100.4 W | 9400 lm | 94 | 2100 K | 24,000 h |
| | 60WP/LED/840/ND EX39 G2 BB 6/1 | 60 W | 8700 lm | 145 | 4000 K | 50,000 h |
| Parking | MH250/U UPC 3PK | 250 W | 21,250 lm | 85 | 4000 K | 10,000 h |
| | 150CC/LED/850/ND EX39 G2 BB 3/1 | 150 W | 24,000 lm | 160 | 5000 K | 50,000 h |

Given that the exterior lights only turn on for the night, the average number of daylight minutes needs to be calculated. The National Oceanic and Atmospheric Administration (NOAA) through the Global Monitoring Laboratory provides a solar calculator to determine the sunrise, sunset, and solar noon along with solar positioning data for the entirety of the Earth [29]. Using the sunlight duration for each day in 2022, an average sunlight duration was calculated. Figure 1 plots the sunlight duration over the course of the year along with the average for the WSUTC campus Floyd parking lot, approximately located at 46.32958° N, 119.26508° W.

With the average daily minutes of sun being 732.2 min for the exterior of the campus, subtracting that value from 1440, the number of minutes in a day, yields the time in which the sun has set. By then taking the number of minutes the sun is set and adding 60 min to it, an accurate average representation of the exterior lighting schedule is created. This 60 min correction is used because the lights are synchronized with a solar timer to come on 30 min prior to sunset and turn off 30 min after sunrise to ensure adequate light is provided. After this correction, the average number of minutes that the lights are is 765.8 min which is approximately 12.76 h. The NOAA calculator determined that the average sunrise and sunset times were 5:49:57 AM and 6:04:09 PM, respectively. To account for the early start and late turn-off, the morning cut-out time is moved to 6:19 AM and evening cut-in time to 6:34 PM. Using Equation (3), the power is obtained for use in Equation (4) and the total energy consumption is calculated. Figure 1 and its relevant data is included to provide an estimate of the average night length. This model does not account for the associated temperature variation.

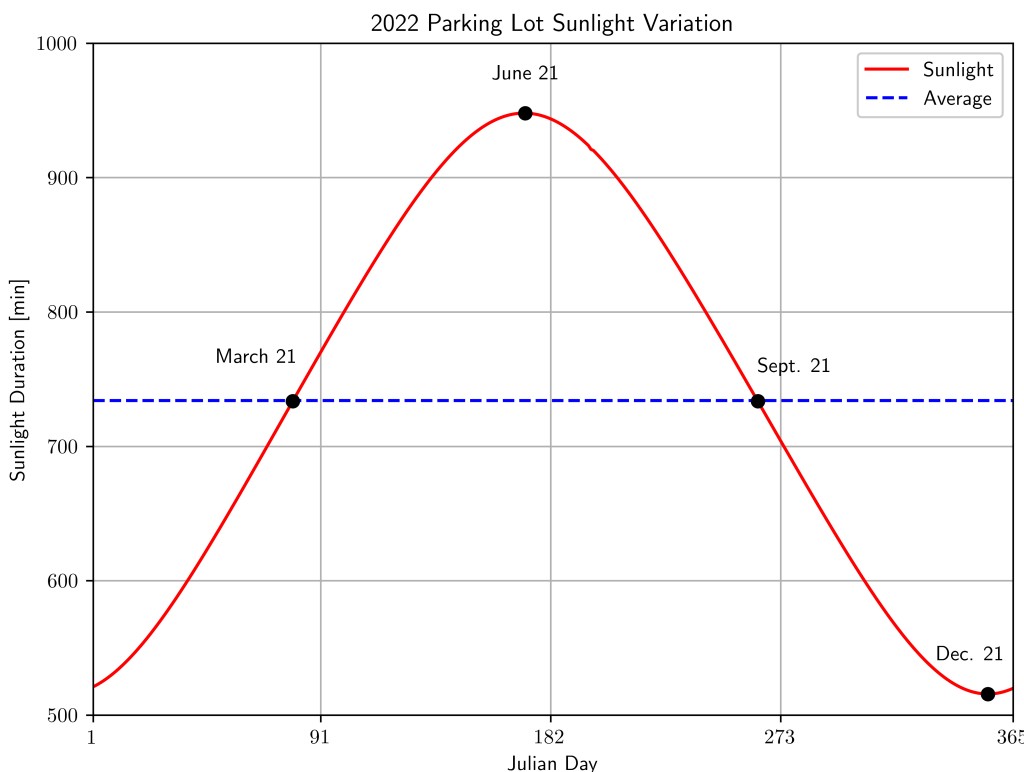

**Figure 1.** Sunlight variation at WSUTC (46.32958° N, 119.26508° W) calculated via NOAA tools with the solsticies and equinoxes plotted as reference.

### 2.3. Classroom and Lab Space

Hallway lighting, stairwell lighting, bathroom lighting, and exterior lighting are fairly easy to analyze, as their light conditions are relatively static and can easily be calculated once the lighting data is available. The problem with classrooms and lab spaces is that each classroom has wildly different occupancy conditions. Since the campus is open for student use, it is not uncommon for students or groups to use the rooms for a quiet place to study between lectures, while the scope of modeling the stochastic nature of students coming in and out of classrooms is out of the scope of this work, the class schedule for each section in a given semester can be used to accurately model light usage strictly in classroom settings. There exists one problem: obtaining this data. Since all of the data is publicly available via an HTML web page, the appropriate fields can be identified and systematically recorded using web crawling tools. In order to achieve the data in a workable format, `BeautifulSoup4 v4.10`, a web crawler library for Python [30] written by Leonard Richardson from New York City, NY, USA, was employed to crawl the publicly-available Spring 2022 semester class data, compiling all classes that meet in-person in the Floyd & East buildings. For each section in these buildings, the start and end times as well as the days classes are held are recorded. Classes start and end with a listed time the being a multiple of 5, starting at 7 AM and ending at 10 PM.

Four cases are presented. The first case only seeks to model the current energy usage, taking in account if there currently exists an occupancy sensor in the room or not. This is included because there are classrooms where retrofits have already been done to add occupancy sensors. If the room already has an occupancy sensor, the lights are modeled to turn off after 5 min of inactivity; thus, between classes, the lights will turn off. If there is no occupancy sensor, the lights are to remain on until the last class of the day in that room. For instance, if a class is held at 9 AM and ends at 10 AM and there is a later class in that room that ends at 5 PM, the lights will be on from 9 AM to 5 PM. The second case then retrofits LED lights into each space using the same occupancy sensing rules as the first case. The third case then adds occupancy to each room without replacing the lights, similar to the

continuous operation group. The final case then combines the retrofit of fluorescent tubes to LED tubes and the inclusion of occupancy sensing.

To model the energy profile, the data for the week is compiled and then averaged out over five days. The power for a given classroom in a given time interval is calculated by:

$$P_j = \frac{1}{5} \sum_{i=1}^{5} p_{ij} \tag{5}$$

where $P_j$ is the average power in the $j$th time interval in the classroom, and $p_{ij}$ is the rated power of the classroom in the $j$th interval for day $i$ in the classroom. $p_{ij}$ is zero if class is not held yet, if occupancy sensors turn the lights off, or if the last class has already ended. Since there are 5 days that classes occur over, $i$ ranges from 1 to 5.

### 2.4. Office & Miscellaneous Lighting

Both faculty and administrative offices require lighting, but their use of lights is not necessarily consistent. For example, a professor may only be in their office for a few hours in the day while the financial aid office is open on a fixed schedule. In addition, there are numerous spaces on-campus that are used on an infrequent basis.

For offices that are open to the public, their office hours shall be used to generate the average daily energy usage. Any offices or work spaces that do not have clearly-defined hours are assumed to be on constantly from 9 AM to 5 PM. The total number of operation hours will be multiplied by the lighting power of that area per Equation (3) on a group-by-group basis, yielding the energy consumed. If the space is designated as being a miscellaneous space used, 3 h of daily usage is used. This approach is taken as opposed to taking occupancy data because of the difficulty of obtaining detailed occupant behavior and to provide a bounding energy consumption in these spaces. In addition, light bulb manufacturers often rate their bulb life based on a daily 3 h usage. As such, only two cases are presented: a no retrofit case and an LED retrofit case.

### 3. Results & Analysis

### 3.1. Lighting Analysis

In Figure 2, the comparison of each retrofit option relative to the base case is presented for the continuous operation group. Table 3 shows the energy consumption data from Figure 2 alongside with the distribution of lights in the category.

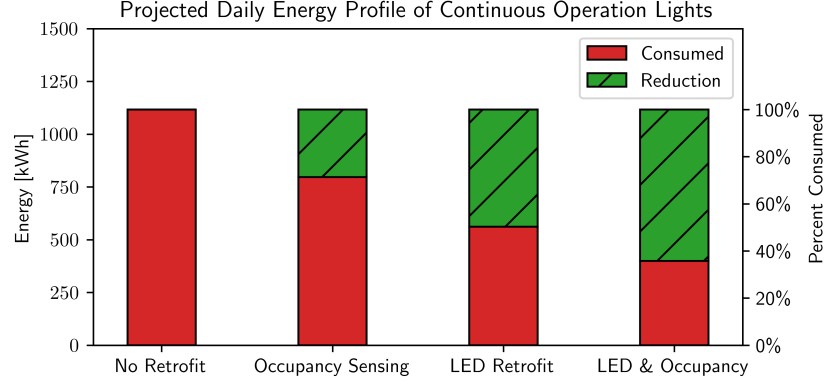

**Figure 2.** Bar chart comparison for continuous operation daily energy usage.

**Table 3.** Continuous operation group light data using a $0.10/kWh utility rate.

| Case | kWh | Energy Cost | Reduction | % Reduction |
|---|---|---|---|---|
| No retrofit | 1117 | $111.70 | - | - |
| Occupancy sensing | 796.8 | $79.68 | $32.02 | 28.7% |
| LED retrofit | 562.2 | $56.22 | $55.48 | 49.7% |
| LED retrofit & occupancy | 399.3 | $39.93 | $71.77 | 64.3% |
| Light Type | Count | Total Power | Egress Lights | Egress Power |
| Fluorescent | 1401 | 44.8 kW | 467 | 14.9 kW |
| LED canister | 171 | 1.7 kW | 0 | 0 kW |

As shown in Figure 2, the LED retrofit had a higher energy reduction than the occupancy sensors. One interpretation of this result is that since the lights are on either the entire day with no occupancy sensor or the majority of the day in the occupancy sensor case, the number of hours in which the lights are off have a less effect in reducing energy than the raw power consumption of the bulbs. Since the lights in the Floyd building have egress lighting and the East building does not, this effect will be further pronounced as lighting upgrades occur in the East building to provide egress lighting. In combining LED lights and occupancy sensors, a 64.3% reduction in energy was observed for this group.

The exterior light results are covered in Table 4, providing information on the type of lights and energy reduction information with Figure 3 showing a visual representation of energy reduction when implementing LED technology. After introduction of LED lights, a reduction of 40% in energy usage is calculated. Given that the exterior lighting analysis uses the same operational hours with different bulbs, these results are expected, as each of the bulbs selected approximately use 40% less energy individually. This is a fairly trivial result; however, these numbers can factored into the overall retrofit profile of the buildings.

**Table 4.** Exterior lighting data using a $0.10/kWh utility rate.

| Case | kWh | Energy Cost | Reduction | % Reduction |
|---|---|---|---|---|
| No retrofit | 154.6 | $15.46 | - | - |
| LED retrofit | 92.7 | $9.27 | $6.19 | 40.0% |
| Light Type | Count | Power | Total Power | |
| Parking lights | 36 | 250 W | 9 kW | |
| Walkway lights | 31 | 100 W | 3.1 kW | |

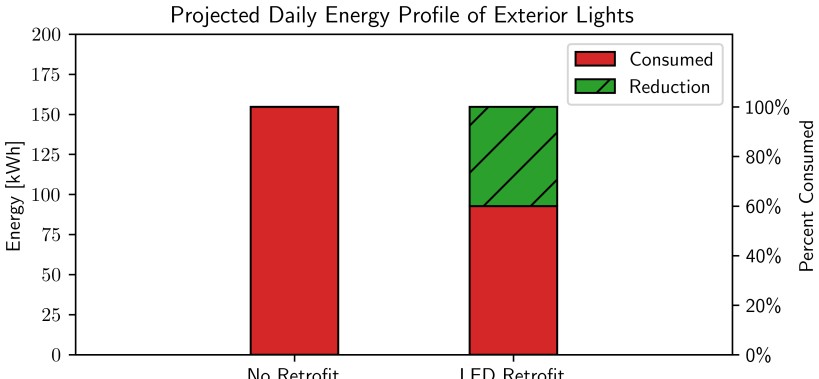

**Figure 3.** Bar chart comparison for exterior lighting daily energy usage.

Using data obtained from crawling the university course catalog for Spring 2022 and combining that with the rules for occupancy as described in Section 2.3, Figure 4 was produced, modeling the variations in power for the average day based on Monday through Friday class data. With no retrofits whatsoever, peak power rates occur from approximately

10 AM to 6 PM with a peak power of just over 14 kW. Both the individual LED and occupancy retrofits were able to shave the peak power down to approximately 8 kW, and the combination of retrofits yielded a max power of approximately 4.5 kW. Figure 5 shows the comparison in energy consumption over the 24 h period modeled with numeric results in Table 5.

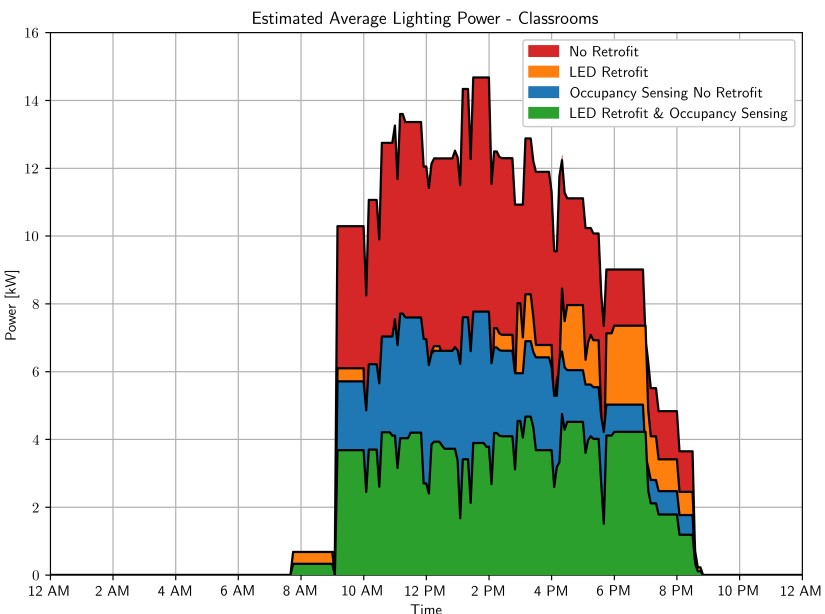

**Figure 4.** Calculated classroom lighting power variation across the average day.

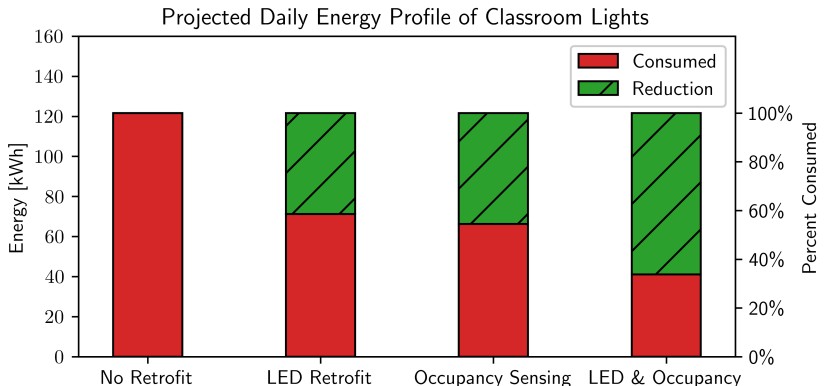

**Figure 5.** Bar chart comparison for daily classroom lighting energy usage.

**Table 5.** Classroom and lab average daily data using a $0.10/kWh utility rate.

| Case | kWh | Energy Cost | Reduction | % Reduction |
|---|---|---|---|---|
| No retrofit | 121.6 | $12.16 | - | - |
| LED retrofit | 71.2 | $7.12 | $5.04 | 41.4% |
| Occupancy sensing | 66.3 | $6.63 | $5.53 | 45.5% |
| LED retrofit & occupancy | 41.1 | $4.11 | $8.05 | 66.2% |

The most notable result present in Table 5 is that the LED retrofit and occupancy sensing were near equally effective in reducing energy consumption relative to the base case, while this result has bias due to the fact that the base case did include occupancy sensors that were already present in the room, the results show a 45.5% reduction in energy consumption via full occupancy sensing and only 41.4% reduction with just LED retrofits. Without counting for occupancy sensors in the no retrofit case, the occupancy sensing

group is expected to show an even larger gap in reduced energy consumption relative to the LED retrofit case as the amount of energy saved by simply turning the lights off can be more readily obtained than the continuous operation group. When combining the two retrofit options, a reduction of 66.2% in energy consumption can be realized, similar to the continuous operation group. Figure 4 shows that the LED retrofit group had lower power consumption during the earlier hours with larger peaks in the evening while the occupancy sensing with no LED retrofit group had the opposite. An explanation for this behavior is that per the methodology, the classes without occupancy sensors will consume power throughout the day as more classes have their lights left on, whereas the occupancy sensors are able to save this power consumption by turning the lights off.

A source of uncertainty in this category is introduced with the assumptions made on how the occupancy sensors are modeled. As described in Section 2.3, rooms without occupancy sensors are modeled as having the lights on from the start of the first class until the end of the last class, effectively modeling the behavior of users not turning the lights out between class sections and that the last class section will turn the lights off without fault at the end of the projected occupancy time for that room. Additionally, all traditional lectures were moved to the Floyd for the Spring 2022 semester due to the COVID-19 pandemic. In general, the classrooms in Floyd contain more rooms that lack occupancy sensors and many sections of the East building were retrofitted with LED lights and occupancy sensors, making the retrofits prior to this study less effective in reducing energy consumption for Spring 2022 because these features are not being utilized over the less equipped classrooms in Floyd. This will tend to increase energy consumption in general across this category.

Figure 6 shows the energy profile of the office & miscellaneous lights with Tables 6 and 7 showing the energy costs and breakdown of lights in each category. Since only LED retrofits were considered in this group, a total energy reduction of 47.4% and 50.6% is calculated from the model for the office and miscellaneous groups, respectively. Because the majority of lights in the office spaces are fluorescent lights that are subject to retrofit, it would follow that their percent reduction would be similar to the reduction of energy between bulbs. Figure 6 shows that this is the case, save for the office spaces having HID lights that influence the power consumption.

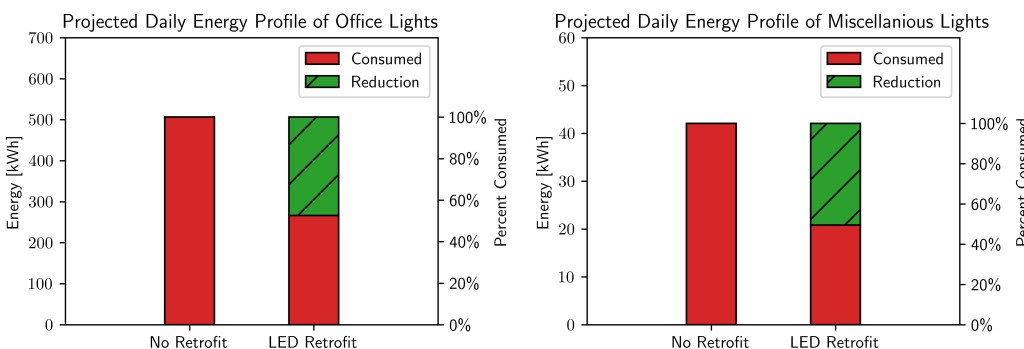

**Figure 6.** Calculated office and miscellaneous light energy consumption.

The office lights have a slightly lower percent reduction than the miscellaneous spaces due to the presence of HID lights that are not being considered for retrofit in this category. Additionally, it should be noted that the numbers presented in Tables 6 and 7 have a degree of uncertainty to them. Based on the assumptions made in Section 2.4, the results will be an overestimation of the energy consumed in these spaces; therefore, the actual reduction in energy consumption will vary. Even with the moderate amount of uncertainty in operational time, the LED retrofit will still be able to provide a similar percentage reduction simply based on the flat reduction in power consumption that is associated with replacing higher power fluorescent bulbs with lower power, more efficient LED bulbs.

**Table 6.** Office group retrofit data at a $0.10/kWh utility rate.

| Case | kWh | Cost | Reduction | % Reduction |
|------|-----|------|-----------|-------------|
| No retrofit | 506.3 | $50.63 | - | - |
| LED retrofit | 266.1 | $26.61 | $24.02 | 47.4% |
| **Light Type** | **Count** | | | |
| Fluorescent | 1820 | | | |
| LED canister | 28 | | | |
| HID | 17 | | | |

**Table 7.** Miscellaneous group retrofit data at a $0.10/kWh utility rate.

| Case | kWh | Cost | Reduction | % Reduction |
|------|-----|------|-----------|-------------|
| No retrofit | 42.1 | $4.21 | - | - |
| LED retrofit | 20.8 | $2.08 | $2.13 | 50.6% |
| **Light Type** | **Count** | | | |
| Fluorescent | 429 | | | |
| LED canister | 30 | | | |

Figure 7 provides a comparison between each retrofit option discussed in this section, providing a visual comparison of energy consumption for each category and retrofit option. Table 8 provides a summary of each retrofit option. For lighting categories that do not have a particular retrofit option, the retrofit option used for calculations is the group without that particular feature. For example, the exterior lights do not have an occupancy sensing group; therefore, the occupancy sensing portion in Figure 7 is the same as the no retrofit category. A key takeaway in Figure 7 is that the continuous operation and office groups consumed more power relative to the rest of the other groups. This shows that retrofits should be prioritized in these groups as they represent the majority of energy consumption in the building for lighting.

**Table 8.** Overall average daily retrofit data for the Floyd & East using a $0.10/kWh utility rate

| Case | kWh | Cost | Reduction | % Reduction |
|------|-----|------|-----------|-------------|
| No retrofit | 1941.7 | $194.17 | - | - |
| Occupancy sensing | 1568.0 | $156.80 | $37.37 | 19.2% |
| LED retrofit | 968.2 | $96.82 | $97.35 | 50.1% |
| LED retrofit & occupancy | 780.0 | $78.00 | $116.14 | 59.8% |

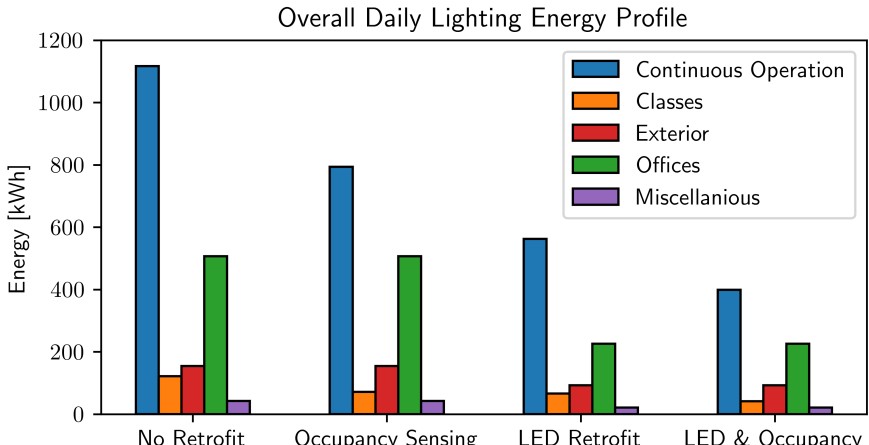

**Figure 7.** Overall retrofit option comparison.

With the model presented in this paper, occupancy sensing is calculated to save 19.2%, LED retrofitting at 50.1%, and LED plus occupancy sensing at 59.8% total energy reduction relative to the no retrofit case, rounding cleanly to 60%. For an estimated 300 days of occupancy, this results in a energy reduction from 582.51 MWh to 234 MWh, totalling approximately 350 MWh in energy savings. At the $0.10/kWh rate, this would translate to an estimated $35,000 in energy savings and a significant energy footprint reduction. Curto, Franzitta, Guercio, and Panno, showed re-lamping with LED lights would generate approximately 100 MWh/year of electrical savings [21], which is similar in magnitude to the results presented in this work.

As mentioned earlier in this section, retrofit options that introduce occupancy sensing are biased to be under performing compared to their LED counterpart. This is a result of not having enough information to account for occupancy in the office and miscellaneous spaces as well as the exterior lights not needing occupancy sensing since they remain on for path lighting and security purposes. Given how in the base retrofit case that the office spaces were the second largest energy consumer and comparable in magnitude to the continuous operation group, the 19.2% reduction is skewed by this category, but without better information, the occupancy sensor energy reduction can be viewed as being near the lower bound of potential energy consumption reduction. Future work is needed to reduce the uncertainty in this number as the potential energy saving with office occupancy sensors are not negligible. The lighting system is also only one portion of the energy portfolio in a building, so work needs to be done to properly assess the mechanical and associated control systems as well as thermal boundaries for potential retrofit options. Additionally, daylight harvesting could be employed to account for the morning and evening sunlight experienced in the atrium spaces. The Floyd building's atrium faces east with a direct line of sight to the horizon; therefore, a fair portion of the morning sunlight can be used in lieu of using any energy at all to operate electric lighting.

### 3.2. Cost Analysis

To determine the break-even point, the cost of the fluorescent and LED bulbs were obtained from online retailers (note that the prices listed may vary and are subject to market forces). The break-even point for the lights is obtained by setting the time-dependent cost of the fluorescent lights equal to the LED lights as shown below in Equation (6). The price for the fluorescent bulb is approximately $6.90/bulb [31] and the LED bulb at £21.75 [32] (approximately $24.72) and a $0.10/kWh utility rate, the time $t$ in hours is solved from:

$$6.9 + \frac{32 \times 0.1}{1000}t = 24.72 + \frac{15.5 \times 0.1}{1000}t \tag{6}$$

Solving the above equation for $t$ yields that the LED and fluorescent lights will break even after 10,800 h of continuous use, or approximately 15 months. This is less than the expected life for both the fluorescent and LED bulbs as reported in Table 1. For the continuous-operation group that operates lights a significant portion of the day, this break-even point will be reached earlier than spaces that use the lights less frequently.

Given the large initial capital cost associated with replacing bulbs, it is suggested that occupancy sensors be placed in high impact areas first, namely the classroom and lab spaces. Only one sensor would be needed for typical classrooms with potentially two for larger classrooms compared to many sets of LED bulbs and the costs associated with properly and safely disposing of fluorescent tubes. After this, LED lights should be installed in the hallways, stairwells, and bathrooms as they represent the single largest energy cost reduction potential. The LED lights used in this study also have a significantly longer life than the currently employed fluorescent lights, resulting in less time spent by facilities to replace failed lights as well as a reduction in hazardous mercury waste generation. The occupancy sensors will need to be properly installed for each space described such that they provide switching capacity commensurate to the methodology.

*3.3. Emissions Reduction*

The 350 MWh of energy savings can be used to calculate an equivalent amount of $CO_2$ emissions reduction using natural gas (methane) as a fuel source. From dimensional analysis, 350 MWh is equivalent to $350 \times 3600$ MJ. According to the American Petroleum Institute (API), natural gas has a volumetric energy density of 37.62 MJ/m$^3$ at standard conditions [33], yielding 33,492.8 m$^3$ of natural gas. API also reports that at standard conditions, there are 23.68 m$^3$ per kgmol [33], and natural gas has a molar mass of 16.04 kg/kgmol. This yields a total of 22,686.8 kg of natural gas. 76.2% of the fuel by mass is comprised of carbon; therefore, using a 1:1 conversion of fuel to $CO_2$ and converting the result to metric tonnes of $CO_2$ shows that the retrofit energy savings can eliminate 62.4 t $CO_2$ emissions per year.

**4. Conclusions**

This paper analyzes the lighting systems at Washington State University Tri-Cities. Five different lighting groups were modeled in this study including: (1) the continuous-operation, (2) exterior, (3) classroom, (4) office, and (5) miscellaneous. In our model, the existing building lights were retrofitted with LED lights and occupancy sensors. The results of this model shows that with both LED retrofitting and occupancy sensing 60% of energy savings, or 350 MWh, in a year was achieved. The 350 MWh energy savings translates into a reduction of 62.4 t of $CO_2$ emissions per year. The results of our cost-analysis in this model shows that the LED light retrofit has a break-even point at 15 months of operation. Future work was also proposed, namely investigating the impacts of light harvesting and to expand the analysis into existing mechanical systems. The analysis presented in this paper assures that the lighting systems can be improved to reduce energy consumption while also providing the occupants of the Floyd & East buildings a comfortable environment to be in. These results also translate well to other locations where the electric energy costs are much larger than $0.10/kWh, such as the state of California or Europe.

**Author Contributions:** Conceptualization, A.P. and M.S.; methodology, A.P.; software, A.P.; validation, A.P.; formal analysis, A.P.; investigation, A.P.; resources, A.P.; data curation, A.P.; writing—original draft preparation, A.P.; writing—review and editing, A.P. and M.S.; visualization, A.P.; supervision, M.S.; project administration, M.S. All authors have read and agreed to the published version of the manuscript.

**Funding:** This research is sponsored by the Washington State University Tri-Cities Campus Facilities Office.

**Institutional Review Board Statement:** Not applicable.

**Informed Consent Statement:** Not applicable.

**Data Availability Statement:** CSV files with lighting information can be found at https://www.kaggle.com/datasets/aronpowers/floyd-east-lighting-dataset?select=Offices.csv (accessed on 13 September 2022).

**Acknowledgments:** The work contained in this article would not have been possible without the specific support and advice of Messiha Saad, Ryan Learn, Randy Slovarp, and Andy Percifield. Additionally, I owe my friends, family, and teachers thanks for their support of me, as I could never have achieved as much without their encouragement.

**Conflicts of Interest:** The authors declares no conflict of interest.

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
