# Peer review of "Building Energy Use: Modeling and Analysis of Lighting Systems—A Case Study"

_sustainability, doi:10.3390/su142013181_

Round 1
Reviewer 1 Report
I recommend accepting this manuscript after improving the quality of the introduction by indicating the importance of the use of LED lamps nowadays as well as its luminescent qualities and lifespan compared to other lamps such as discharge lamps, fluorescent,..... In addition, the introduction must be enriched by new publications interested in LEDs. For example, the following references:
1/Optimization of fins arrangements for the square light emitting diode (LED) cooling through nanofluid-filled microchannel, Scientific Reports, 11, 12610 (2021).
2/A three-dimensional thermal analysis for cooling a square Light Emitting Diode by Multiwalled Carbon Nanotube-nanofluid-filled in a rectangular microchannel, Advances in Mechanical Engineering, Vol. 13(11) 1–14, (2021).
3/A three-dimensional thermal management study for cooling a square Light Edding Diode, Case Studies in Thermal Engineering, 101223, 2021.
4/A three-dimensional thermal analysis and optimization of square light edding diode subcomponents, International Communication in Heat and Mass Transfer, 105016, (2020).
5/Heat transfer enhancement of circular and square LED geometry, International journal of numerical methods for heat and fluid flow, Vol. 29 No. 5, pp. 1877-1898, (2019)
6/Alternate PCM with air cavities in LED heat sink for transient thermal, International journal of numerical methods for heat and fluid flow, Vol. 29 No. 10, (2019).
Reviewer 2 Report
Building Energy Analysis: A Case Study Analyzing Potential Lighting Retrofits in American University Buildings
This article describes an essential field that needs to be studied in order to develop novel energy-saving strategies. The authors have contributed a reasonable effort to gather information regarding the topic. However, I would like to point out a few errors that would enhance the quality of the manuscript.
1) I suggest the authors include an introduction explaining the importance of studying the building energy analysis by giving some examples of energy over usage and the possible outcomes that can happen if the energy is overly used.
2) I suggest the authors consider revising the title because the paper entity describes mostly the Washington state building system, not the entire American University buildings.
3) Under section 1.2 (Impact of Washington State law), I suggest removing these sentences “After three revisions, the bill passed the Senate on April 15th and the House on April 18th. It was then signed into law by Governor Jay Inslee on May 7th, 2019 with an effective date of July 28th that same year”
Because these additional sentences bring the manuscript to a story feeling which deviated from the scientific scale.
4) I suggest the authors include a graphical abstract to enhance the attention of the reader.
Reviewer 3 Report
The authors have made an interesting study on Building Energy Analysis: A Case Study Analyzing Potential Lighting Retrofits in American University Buildings. The manuscript can be published but the authors need to justify the scientific writing manuscript. Some of the general comments are provided below:
1. Abstract is lacking the main findings of the work.
2. The work is based on the theoretical calculation of the light power consumed in the building and lacks information on losses associated with the lightning system.
3. The details design analysis of the building affecting the lightning system in the building is not considered in the current study (windows, glass walls etc)
4. The work is lacking experimental work, would be better to calculate the power consumption of the building through experimentation and verify the theoretical equation suggested for the study.
5. The study is also lacking the economic analysis of the Lighting Retrofits (Break-even point etc).
6. The work is lacking novelty as the use of motion sensor-based technology is already available in most parts of the world and is established fact that it can be used for power saving.
7. The work can be improved by incorporating the emission reduction model for the current case study.
Reviewer 4 Report
The manuscript with the title “Building Energy Analysis: A Case Study Analyzing Potential Lighting Retrofits in American University Buildings” focused on the lighting systems at Washington State University Tri-Cities, accounting for several different groups of lighting and providing retrofit options accordingly. The manuscript is technically well written. Below are my comments:
1. Did the author consider the energy consumed by an instrument or device which was later removed or replaced during the data analysis period? If it was done, then how did it affect energy consumption?
2. Figure 1, the duration of sunlight was calculated. How about the temperature variation during the year and how it influences the final result?
3. Except for the energy-efficient bulbs (LED, Fluorescent), what were the other things required in order to retrofit these bulbs?
4. As mentioned in the manuscript, the bulbs are costly. How much it will save if we compare its energy-saving capacity to its cost?
Reviewer 5 Report
The paper is devoted for analysis the lighting systems analysis in some Washington university buildings. However, the paper contains unexplained places and need major revisions.
The aim of the paper should be more clearly formulated.
Introduction should be expanded and more useful references should be added. Obtained results should be more compared with results already published in literature.
Results presented in Figs. 2-7 should be more discussed.
Conclusions should be rewritten in more informative way.
Numbers and measurements units should be written separately. For example line 141 ‘’60 W’’, not ‘’60W’’.
Round 2
Reviewer 3 Report
The authors have modified the article, so it is fine for publication now.
Reviewer 5 Report
Authors make proper corrections according to reviewer remarks and I suggest to publish the paper as it is.